# Hydrophobin Gene *Cmhyd4* Negatively Regulates Fruiting Body Development in Edible Fungi *Cordyceps militaris*

**DOI:** 10.3390/ijms24054586

**Published:** 2023-02-27

**Authors:** Xiao Li, Mengqian Liu, Caihong Dong

**Affiliations:** 1State Key Laboratory of Mycology, Institute of Microbiology, Chinese Academy of Sciences, Beijing 100101, China; 2College of Horticulture, Hebei Agricultural University, Baoding 071001, China; 3University of Chinese Academy of Sciences, Beijing 101408, China

**Keywords:** *Cordyceps militaris*, hydrophobin, *Cmhyd4*, negatively regulate, fruiting body development

## Abstract

A deep understanding of the mechanism of fruiting body development is important for mushroom breeding and cultivation. Hydrophobins, small proteins exclusively secreted by fungi, have been proven to regulate the fruiting body development in many macro fungi. In this study, the hydrophobin gene *Cmhyd4* was revealed to negatively regulate the fruiting body development in *Cordyceps militaris*, a famous edible and medicinal mushroom. Neither the overexpression nor the deletion of *Cmhyd4* affected the mycelial growth rate, the hydrophobicity of the mycelia and conidia, or the conidial virulence on silkworm pupae. There was also no difference between the micromorphology of the hyphae and conidia in WT and Δ*Cmhyd4* strains observed by SEM. However, the Δ*Cmhyd4* strain showed thicker aerial mycelia in darkness and quicker growth rates under abiotic stress than the WT strain. The deletion of *Cmhyd4* could promote conidia production and increase the contents of carotenoid and adenosine. The biological efficiency of the fruiting body was remarkably increased in the Δ*Cmhyd4* strain compared with the WT strain by improving the fruiting body density, not the height. It was indicated that *Cmhyd4* played a negative role in fruiting body development. These results revealed that the diverse negative roles and regulatory effects of *Cmhyd4* were totally different from those of *Cmhyd1* in *C. militaris* and provided insights into the developmental regulatory mechanism of *C. militaris* and candidate genes for *C. militaris* strain breeding.

## 1. Introduction

Filamentous fungi have formed molecular adaptations to the physicochemical challenges related to their lifestyle after one billion years of evolution [1]. For instance, filamentous fungi commonly secrete small amphiphilic and highly surface-active protein hydrophobins [2,3]. They are initially secreted in a soluble form, spontaneously localize at the hydrophilic/hydrophobic interface, and then are assembled into amphipathic layers of varying solubility [4,5].

Hydrophobin layers significantly reduce the interfacial tension, decrease wetting, and provide the ability to adapt to environmental pressures, thus letting the hyphae break through the surface of the liquid and grow into the air by forming aerial hyphae, spores, and even fruiting bodies in macro fungi [6,7,8]. In *Schizophyllum commune* (Agaricales), the transcripts of the hydrophobins *SC3* and *SC4* markedly increase in vegetative hyphae, mounds, and fruiting bodies [9]. The inactivation of the hydrophobin gene *Hyd9* results in a decrease in aerial mycelia and primordia formation in *Flammulina filiformis* (Agaricales) [7]. Moreover, hydrophobin can encourage both spores and hyphae to adhere to hydrophobic surfaces or interact with symbiotic partners [10,11,12,13] and influence the growth and development of pathogenic fungi [14,15,16]. Luciano-Rosario et al. [13] reported that during competitive pathogenicity tests on apples, the hydrophobin septuple-deletion mutant in *Penicillium expansum* (Eurotiales) showed a decrease in spore dispersal and was more suitable than the wild type. In the entomopathogenic fungus *Beauveria bassiana* (Hypocreales), *hyd1* was found to positively regulate the hydrophobicity and fungal virulence, while *hyd2* played a positive role in the surface hydrophobicity of conidia without affecting virulence [17]. In another entomopathogenic (as well as macro) fungus, *C. militaris* (Hypocreales), the hydrophobin *Cmhyd1* showed increased transcription during fruiting body development compared with the mycelial stage and was proven to play a positive role in conidiation, infection, and primordium differentiation [18,19].

*C. militaris*, a model organism for the study of *Cordyceps* spp., can form fruiting bodies on both silkworm pupae and wheat media [20,21]. Its fruiting bodies have been used as food and nutrition tonics worldwide, especially in eastern Asia, due to their various effective constituents, such as carotenoids, adenosine, cordycepin, mannitol, and pentostatin [22,23]. *C. militaris* has become a research model for the growth and development of edible fungi, owing to its genetic tractability and controllable asexual and sexual developments. The fruiting body formation was demonstrated to be regulated by the photoreceptor genes *Cmwc-1* and *Cmvvd*, as well as the hydrophobin gene *Cmhyd1* [18,19,24,25].

Our previous study found that there existed four hydrophobin-encoding genes in *C. militaris* [19]. There were different transcript patterns among the four hydrophobin genes during fruiting body development, indicating their different roles [19]. *Cmhyd4* was found to be only transcribed at the mycelial stage in the WT strain cultured in darkness, with very little transcription after light exposure, pupae infection, and fruiting body development [19]. In this study, the negative regulation of the fruiting body development of a hydrophobin gene was verified by reverse genetics in the edible fungus *C. militaris*.

## 2. Results

### 2.1. Deletion, Complementation, and Overexpression of Cmhyd4 in Cordyceps militaris

The *Cmhyd4* knockout (Δ*Cmhyd4*) strain was generated by homologous recombination (Figure 1A). Genetic complementation (Δ*Cmhyd4c*) and overexpression (*Cmhyd4oe*) strains were produced by introducing *Cmhyd4* with the original promoter of *Cmhyd4* and a strong promoter of *Cmgpda* to *Cmhyd4* disruption and wild-type (WT) strains, respectively (Figure 1).

Gene-disruption and genetic-complementation strains of *Cmhyd4* were initially identified by PCR with the primers listed in Appendix A (Figure 1B, Appendix A), then confirmed by Southern blot assay with an amplified probe, as well as RT-PCR. The WT and Δ*Cmhyd4* strains showed 1.61-kb and 2.67-kb hybridized bands, respectively, whereas 1.61-kb and 2.67-kb hybridized bands were presented by the Δ*Cmhyd4c* strains (Figure 1D). The RT-PCR results revealed that the transcription of *Cmhyd4* was inactive in the Δ*Cmhyd4* strain, the same as that in the WT strain in the two complementation strains (Δ*Cmhyd4c-1* and Δ*Cmhyd4c-2*), and about 3.0–4.0-fold that in the WT strain in the two *Cmhyd4oe* strains (*Cmhyd4oe-4* and *Cmhyd4oe-5*) (Figure 1C). One gene-disruption strain (Δ*Cmhyd4*), two gene-complementation strains (Δ*Cmhyd4c-1* and Δ*Cmhyd4c-2*), and two gene-overexpression strains (*Cmhyd4oe-4* and *Cmhyd4oe-5*) of *Cmhyd4* were obtained and used for the phenotype observation.

### 2.2. Deletion of Cmhyd4 Affects the Aerial mycelia in Darkness and the Colony Color after Light Irradiation

When the WT, Δ*Cmhyd4*, *Cmhyd4c*, and *Cmhyd4oe* strains were cultured in a test tube with PDA in darkness, the aerial mycelia of the Δ*Cmhyd4* strain were the thickest among all the strains (Figure 2A). All the strains, cultured on PDA plates after light irradiation for 4 d, showed obvious differences in colony color. The colony color of the Δ*Cmhyd4* strain was deeper than that of the WT strain. The color of the Δ*Cmhyd4c* culture was saffron yellow, similar to that of the WT, while the color of the *Cmhyd4oe* culture was almost white (Figure 2B). These observations implied that *Cmhyd4* could be correlated to pigment synthesis in *C. militaris*.

The growth rates on PDA medium (Figure 2B,E) in the *Cmhyd4* mutant strains, as well as the hydrophobicity of the aerial mycelia, stayed the same as in the WT strain (Figure 2C). There was no distinct difference between the hyphal morphology of the Δ*Cmhyd4* and WT strains according to scanning electron microscopy (SEM) (Figure 3A,B).

The Δ*Cmhyd4* strain showed a significantly increased conidial yield (*p* < 0.05) compared with that of the WT strain (Figure 2F). The conidial production of the two complementation strains Δ*Cmhyd4c-1* and Δ*Cmhyd4c-2* was similar to that of the WT strain (Figure 2F), while in the two overexpression strains *Cmhyd4oe-4* and *Cmhyd4oe-5*, there was a significant decrease compared to the WT strain (Figure 2F). However, the conidia of the Δ*Cmhyd4* strain were well-dispersed in 0.1% Tween 80, implying that the deletion of *Cmhyd4* did not affect the hydrophobicity of the conidia (Figure 2D). On the surface of the conidia, the granular attachments of the Δ*Cmhyd4* strain were also the same as those of the WT strain observed by SEM (Figure 3A,B).

### 2.3. Deletion of Cmhyd4 Affects Carotenoid and Adenosine Production in Cordyceps militaris

Our previous report in *C. militaris* found that two-stage culture could effectively increase the natural carotenoid content [26]. Therefore, we determined the secondary metabolite production of the WT, Δ*Cmhyd1*, and Δ*Cmhyd4* strains by two-stage cultures. When *C. militaris* was cultured in potato dextrose broth (PDB) in flasks at 150 r.p.m. for 4 d and then statically incubated for another 14 d under a 12 h:12 h light/dark cycle, the vela were formed. The velum color of both the upper and reverse sides in the Δ*Cmhyd4* strain was the deepest among all the tested strains (Figure 3A). The Δ*Cmhyd1* strain appeared white for both the upper and reverse sides of the velum. The carotenoid content of the Δ*Cmhyd4* strain was two-fold higher than that of WT strain, and there was little carotenoid in the Δ*Cmhyd1* strain, which was consistent with the morphology observations (Figure 4A,B).

In *C. militaris*, both cordycepin and adenosine are important active compounds [27,28]. The Δ*Cmhyd4* strain showed a similar cordycepin content to the WT strain and a higher content than that of the Δ*Cmhyd1* strain, while the adenosine content increased two-fold in the absence of *Cmhyd4* and eight-fold in the absence of *Cmhyd1* compared with the WT strain (Figure 4B).

### 2.4. Deletion of Cmhyd4 Affects Stress Responses in Cordyceps militaris

Compared with the WT and complementation strains, the mycelia of the Δ*Cmhyd4* strain grew more quickly in the presence of 10% CR, 0.1% SDS, and 0.075 mM H_2_O_2_, especially in the presence of 0.075 mM H_2_O_2_ (Figure 5). It was inferred that the tolerance of the mycelia to oxidant stress was affected after *Cmhyd4* deletion.

### 2.5. Cmhyd4 Plays no Role in Cuticle-Bypassing Infection in Pupae

In the field or by artificial cultivation, the club-like fruiting bodies are formed after the conidia of *C. militaris* infecting pupae. It was found that *Cmhyd1* showed a crucial role in cuticle-bypassing infection [18]. However, when the *Cmhyd4* mutant strains inoculated the silkworm pupae by infection, there was no difference in the days for mummification between all the mutants and the WT strain (Figure 6), implying that *Cmhyd4* played no role in cuticle-bypassing infection.

### 2.6. Cmhyd4 Negatively Regulates Fruiting Body Development in Cordyceps militaris

*C. militaris* is a well-known edible and medicinal mushroom, and the fruiting body is the main edible part. The Δ*Cmhyd4* and Δ*Cmhyd4c* strains could form primordia and normal fruiting bodies, while the *Cmhyd4oe* strains developed malformed fruiting bodies (Figure 7A). Moreover, the biological efficiency of the Δ*Cmhyd4* strain was significantly increased compared to that of the WT strain. Further analysis found that the deletion of *Cmhyd4* increased the biological conversion rate by improving the fruiting body density, not the height (Figure 7B).

Our previous study showed that the MAPK signaling pathway played a role in fruiting body development and formation in *C. militaris* [18]. The transcription levels of *Cmhyd4* and seven genes in the MAPK signaling pathway, namely, the Pth11-like G-protein-coupled receptor gene (*Pth11,* CCM_03015); the Ras small GTPase gene (*CDC42*, CCM_00979); *STE7* (MAPKK, CCM_03428); *STE11* (MAPKKK, CCM_02296); *STE20* (MAPKKKK, CCM_09268); and the mitogen-activated protein kinase gene (*ERK*, CCM_01235, CCM_09637) [18,29], were compared between the stages of mycelia and primordia in the WT and Δ*Cmhyd4* strains. At the stage of the primordium, *Cmhyd4* showed a very low transcription level compared with the mycelial stage in the WT strain (Figure 7C). Compared with the mycelial stage in the WT strain, the transcription of all the tested genes in the MAPK signaling pathway except *ERK1* and *ERK2* experienced a significant upregulation at the primordium stage, which was consistent with our previous results [18]. In the Δ*Cmhyd4* strain, the transcription levels of *Pth11*, *CDC42*, *STE7*, *STE11*, and *STE20* at the primordial stage exhibited a similar upregulation compared to the WT strain (Figure 7C, Appendix A Appendix A).

## 3. Discussion

In *C. militaris*, four hydrophobin-encoding genes have been recovered. The class Ⅱ hydrophobin *Cmhyd1*, expressed highly during fruiting body formation and development compared with the mycelial stage, was confirmed to positively regulate aerial mycelial growth, conidia production, the hydrophobicity of the mycelia and conidia, pigment synthesis, pupae infection, and primordium formation [18,19]. The class Ⅰ hydrophobin *Cmhyd4* only presented a high transcription level at the mycelial stage under dark condition [19], implying a different biological function from *Cmhyd1* in *C. militaris*. In this study, it was found that *Cmhyd4* negatively regulated aerial mycelial growth; conidiation; carotenoid and adenosine synthesis; resistance to oxidant stress, and fruiting body development by gene deletion, overexpression, and complementation.

Studies in pathogenic fungi have indicated that hydrophobins may be important in fungal–host interactions [17,30]. In the human pathogenic fungus *Aspergillus fumigatus*, hydrophobin *rodA* was reported to regulate the rodlet structure of conidia, play an appreciable role in insect hemocyte immune evasion, and be involved in the resistance of conidia to host cells [31]. In the plant pathogenic fungus *Magnaporthe grisea* (Magnaporthales), *MHP1* has a key role in hydrophobicity and infection-related fungal development and is a requisite for pathogenicity [32]. In the entomogenous fungus *Beauveria bassiana* (Hypocreales), it has been reported that both *hyd1* and *hyd2* play no appreciable role in insect hemocyte immune evasion, despite their effects on conidia hydrophobicity and rodlet layers [17]. Δ*Cmhyd1* strains in the entomogenous fungus *C. militaris* exhibited lower infection rates than the WT strain over the whole silkworm pupae infection period via cuticle-bypassing infection [18]. However, there was no difference between the infection ability in pupae between the WT and Δ*Cmhyd4* strains in this study. Moreover, the deletion of *Cmhyd4*, in contrast to the deletion of *Cmhyd1*, had no influence on the hydrophobicity or granular attachments on the conidial surface (Figure 2C, 2D; Figure 3). Although there were no rodlets observed on the outer surface of the conidia in the WT, Δ*Cmhyd1*, and Δ*Cmhyd4* strains, it is possible that the hydrophobin in *C. militaris* regulated the conidial virulence by affecting granular attachments on the conidial surface. Alternatively, the combined effects of conidial clumping (lowered germination) could have been due to the lower hydrophobicity and lowered hyphal growth rate in Δ*Cmhyd1*. However, it should be noted that the silkworm pupae could not be mummified completely after being infected with the conidia of Δ*Cmhyd1* via cuticle-bypassing infection but were mummified completely after adding heterogenous expression to CmHYD1. Thus, the hydrophobins in *C. militaris* play an appreciable role in insect hemocyte immune evasion, unlike the hydrophobins in *B. bassiana* [17], implying the involvement of birth-and-death-type hydrophobin gene family evolution, resulting in functionally diverse proteins.

Most of the functional genes affected fruiting body development and the vegetative growth rate. For example, the hydrophobin CmHYD1 and photoreceptor CmWC-1 in *C. militaris* [18,24]; the hydrophobin Hyd9 [7], Gβ-like protein FvCPC2 [33], and PDD1 in *F. filiformis* [34]; and GlSwi6 in *Ganoderma lucidum* [35] affected both fruiting body development and the vegetative growth rate. Other genes have opposite roles in fruiting body development and the vegetative growth rate. The transcription factors *Fst4*, *Hom2*, *Tea1*, and *WC-2* in *S. commune* repressed the vegetative growth rate but promoted fruiting body development, while *Hom1* promoted the vegetative growth rate but suppressed fruiting body development [36,37,38]. As far as we know, there are few genes that specifically regulate fruiting body development, and they have no influence on the vegetative growth rate—these include *fst3* and *gat1* in *S. commune* [36,37,38] and *lfc1* in *F. filiformis* [39]. In this study, the hydrophobin *Cmhyd4* belonged to this class of functional genes.

Regarding the transcription regulation of hydrophobins, there are very few reports on the direct regulation of upstream gene transcription, and most research stops at the level of gene expression. During hydrophobin synthesis, the nutrition-linked regulatory mechanism in *Ma. grisea* [40,41], *Metarhizium anisopliae* [42], and *Me. robertsii* [43,44] may regulate the transcription of hydrophobin genes. A previous study in our laboratory showed that the nitrogen regulation transcription factor CmAreA was recruited to the promoter of *Cmhyd1* and activated the transcription of *Cmhyd1* with the coactivation of CmOTam [18]. In *Trichoderma guizhouense*, hydrophobin production was related to blue light receptors [45]. We confirmed that the transcription of *Cmhyd1* was not regulated by light and photoreceptors, but the transcription of *Cmhyd4* was regulated negatively by light and the photoreceptor *Cmwc-1* [19]. We tried to verify whether CmWC-1 could directly bind the promoter of *Cmhyd4* but failed. In our latest study involving ChIP-seq, a 12 bp region AAATCAGACCAC/GTGGTCTGATTT, predominated by AAATCA in positions 1–6 and CCAC in positions 9–12, was identified as a binding site for CmWC-1. There was no binding site in the promoter of *Cmhyd4*, and *Cmhyd4* was not among the 427 target genes directly regulated by CmWC-1 [46]. *Cmwc-1* was assumed to regulate the transcription of *Cmhyd4* by indirect binding and, possibly, other unknown factors.

After *Cmhyd4* deletion, the density and biological conversion rate of the fruiting body were remarkably increased compared with the WT strain (Figure 7). In addition, the carotenoid and adenosine contents also increased one- and two-fold compared to those of the WT strain (Figure 4). When a mushroom cultivar has these features, it presents a higher yield with more nutritional contents. Moreover, highly efficient marker-free gene editing by the effective AMA1-based CRISPR/Cas9 system in *C. militaris* has been realized [47]. In the future, *Cmhyd4* could be useful as a target gene for *C. militaris* strain breeding.

## 4. Materials and Methods

### 4.1. Strains and Culture Conditions

The *C. militaris* strain (No. CGMCC 3.16323) was stored on potato dextrose agar (PDA) at 20 °C in darkness. The *Escherichia coli* strain DH5α and *Agrobacterium tumefaciens* strain AGL-1 (Tiangen Biotech Co., Ltd., Beijing, China) were maintained at −80 °C and cultured with Luria Bertani medium supplemented with kanamycin (50 μg/mL) and yeast extract beef (YEB) medium supplemented with carbenicillin (50 μg/mL) and kanamycin (50 μg/mL), respectively, for resistant label screening.

During phenotype observation, the *C. militaris* strains were cultured in constant darkness at 20 °C for 21 d and then placed under dark/light (12 h/12 h) condition at 20 °C for 4 d. After the colonies were cultured on PDA medium for 21 d, the diameters of the colony were measured to evaluate the growth rates by the cross-line method. The conidial production of each colony was calculated from six replicate plates with three counts for each strain after exposure to dark/light (12 h/12 h) condition at 20 °C for 4 d. Liquid cultures were performed in media containing 200 g/L potato, 20 g/L glucose, 3 g/L peptone, 1 g/L KH_2_PO_4_, and 0.5 g/L MgSO_4_ as two-stage cultures.

### 4.2. Gene Deletion, Complementation, and Overexpression

According to the *C. militaris* strain CM01 genome (accession: SRA047932), the ORF cassette of *Cmhyd4* with 1381 bp upstream and 1328 bp downstream was obtained [29]. Based on vector pAg1-H3 (provided by Prof. Xingzhong Liu from Nankai University), the deletion vectors pAg1-hyg-*Cmhyd4* were cloned with upstream hygromycin-resistance gene (*Hyg*) cassette and downstream insertion. After *A. tumefaciens*-mediated transformation, mutant strains were obtained and verified by polymerase chain reaction (PCR), Southern blot, and reverse transcription PCR (RT-PCR).

For gene complementation, the 1, 226 bp *Cmhyd4*-coding sequence containing its putative promoter and terminator regions was amplified from the WT strain and ligated into vector pAg1-bar (this lab) to generate pAg1-bar-*Cmhyd4*. The pAg1-bar-*Cmhyd4* was introduced into the Δ*Cmhyd4* strain. The *Cmhyd4* complementation mutants were screened with 2 mg/mL glufosinate (bar) and then verified by PCR, Southern blot, and RT-PCR.

*Cmhyd4* overexpression vector pAg1-hyg-P*_Cmgpda_*-*Cmhyd4* was obtained with the strong promoter *Cmgpda* from *C. militaris*, the 374 bp *Cmhyd4* ORF sequence, and the terminator TtrpC from *A. nidulans*, which were inserted into pAg1-H3. The *Cmhyd4* overexpression strains were screened and verified by PCR and RT-PCR. All the vector constructions were generated with a CloneExpress^®^ Ultra One Step Cloning Kit (C115, Vazyme Biotech Co., Ltd., Beijing, China). Appendix A lists the primer sequences used in this study.

### 4.3. Hydrophobicity Determination for the Mycelia and Conidia

At the surface of the mycelia, we added 20 μL of 0.1% Tween 80 and observed the diffusion capacity for 30 min to check their hydrophobicity. As for the conidial hydrophobicity, we observed the conidial suspensions containing 0.1% Tween 80 solution using a light microscope for 30 min.

### 4.4. Hyphae and Conidia Observation by Scanning Electron Microscopy

Under 20 °C condition, the WT and all mutant strains were grown in constant darkness for 21 d and transferred to dark/light (12/12 h) condition for 2 d. We observed the hyphae and conidia using a scanning electron microscope (SEM) (Hitachi SU8010, Tokyo, Japan). The experimental procedures were described in detail in our previous study [19].

### 4.5. Determination of Carotenoid, Cordycepin, and Adenosine Contents

The WT, Δ*Cmhyd1*, and Δ*Cmhyd4* strains were cultured in PDB using a two-stage culture method. Firstly, we conducted the culture at 150 r.p.m. in darkness at 20 °C for 4 d. Secondly, we transferred the flasks to an illumination incubator (MGC-450BP, Yiheng Instruments, Shanghai, China) at 20 °C in light for 14 d for velum forming. We collected and dried the velums and then determined the carotenoid content using the method we published in [48]. Cordycepin and adenosine were quantified by high-performance liquid chromatography, as previously published in [49].

### 4.6. Assays for Stress Adaptation

We determined the effects of osmotic and oxidative stresses on growth by the method described by Li et al. [18]. After culturing in constant darkness at 20 °C for 21 d, the colony was photographed.

### 4.7. Assays for Fungal Virulence in Silkworm Pupae

For cuticle-bypassing infection, a 100 μL conidia suspension at 10^7^/mL was injected into each silkworm pupa in three groups (6 pupae per group). All treated pupae were maintained at 20 °C, and we monitored the conidial infection rate daily until all the pupae were mummified completely.

### 4.8. Fruiting Body Formation in Cordyceps militaris

We cultivated the strains for fruiting in wheat medium according to our previously described method [19]. The fresh weights of the fruiting bodies were recorded after harvest. The formula for calculating the biological efficiency (BE) of each strain was BE (%) = (weight of fresh stroma/weight of dry substrate) × 100. The biological efficiency, number, and height of the fruiting bodies in each vessel were measured among the three groups (5 vessels per group). These fruiting experiments were conducted three times independently.

### 4.9. RNA Extraction and RT-qPCR Analysis

An E.Z.N.A.™ Plant RNA Kit (Omega, Stamford, CT, USA) was used to isolate the total RNA from frozen mycelia or fruiting bodies. A HiScript III 1st Strand cDNA Synthesis Kit (+gDNA wiper) (Vazyme Biotech Co., Ltd., Beijing, China) was used to synthesize the first-strand cDNA from the total RNA. RT-qPCR was conducted using a CFX Connect Real-Time System (Bio-Rad, Singapore) with 10 µL qPCR solutions containing 5 μL AceQ^®^ qPCR SYBR^®^ Green Master Mix, 2 ng cDNA, and 0.1 μM primers. The *rpb1* (CCM_05485) was used as an internal standard [50]. The 2^−ΔΔCt^ method was used to calculate the relative gene expression levels [51]. Appendix A lists all the primer sequences used in this study. The data provided represent three biological replicates and three technical replicates.

## 5. Conclusions

This study revealed negative roles for the novel putative hydrophobin gene *Cmhyd4* in aerial mycelia, mycelia stress resistance, conidiation, secondary metabolites, and primordium formation in *C. militaris*. Furthermore, *Cmhyd4* had no effects on the mycelial growth rate, hydrophobicity, microphenotype of hyphae and conidia or conidial virulence in silkworm pupae. Since the deletion of *Cmhyd4* could promote fruiting body development and increase yields, CmHYD4 and its homologous encoding genes could be useful in *C. militaris* breeding.

## Figures and Tables

**Figure 1 ijms-24-04586-f001:**
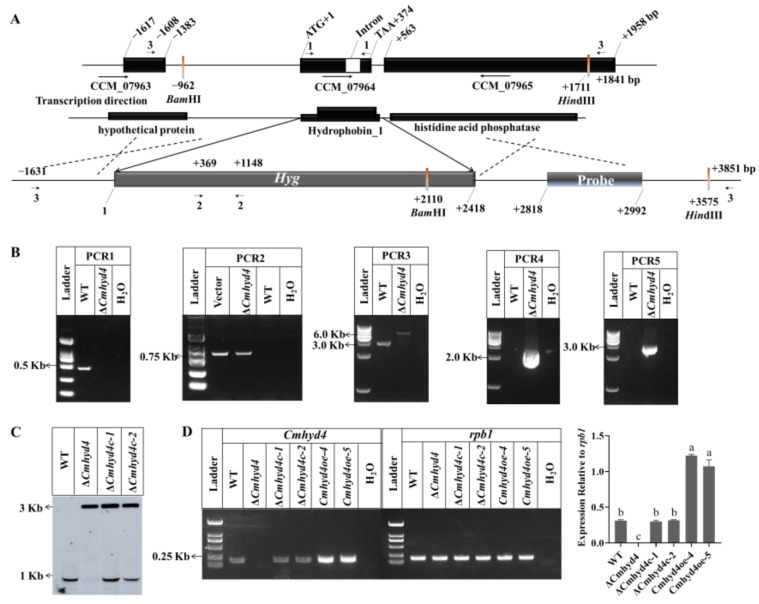
Generation and identification of the *Cmhyd4* mutants. (**A**) Scheme for the Δ*Cmhyd4* strain obtained via homologous recombination; *Cmhyd4* was replaced by the hygromycin resistance gene. (**B**) Confirmation of *Cmhyd4* deletion by PCR. The primers 1, 2, and 3 were used to perform PCR1, PCR2, and PCR3, respectively; PCR4 and PCR5 were performed with the primers 3F/2R and 2F/3R, respectively. (**C**) Southern blot assay with double-digest sites *BamH*I and *Hind*III validating the *Cmhyd4* gene deletion and complementation strains. Lane 1: WT; lane 2: Δ*Cmhyd4* strain; lanes 3 and 4: Δ*Cmhyd4c-1* and Δ*Cmhyd4c-2* strains, respectively. (**D**) The transcription levels of *Cmhyd4* in the WT and *Cmhyd4* mutants according to RT-PCR. The transcription levels of *Cmhyd4* relative to *rpb1* are presented as the optical density ratio obtained using ImageJ v.1, and the different letters over the histogram indicate significant differences at *p* < 0.05.

**Figure 2 ijms-24-04586-f002:**
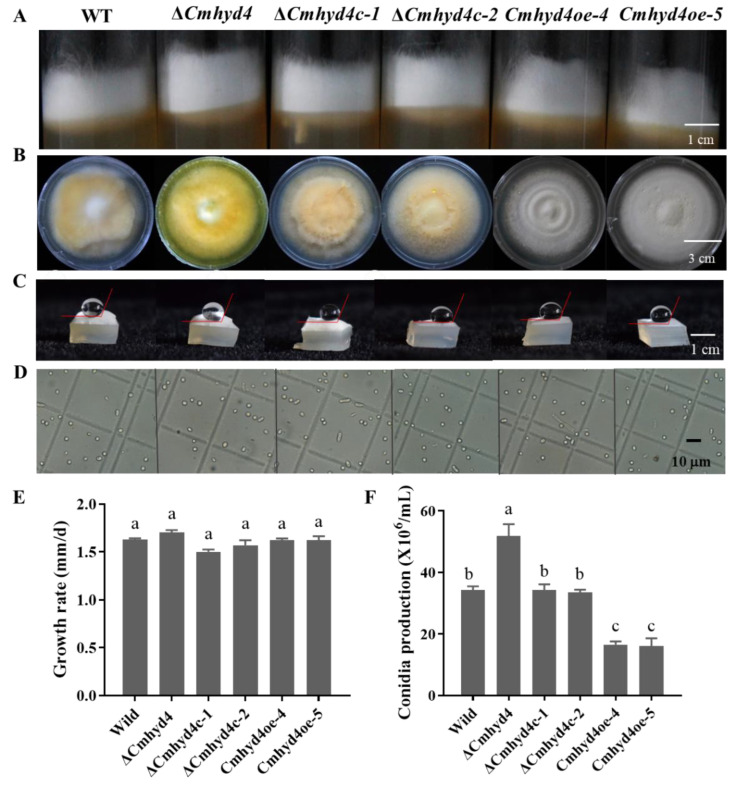
Morphology, growth rate, hydrophobicity, and conidiation in the WT and mutant strains. (**A**) Aerial mycelia: Aerial mycelia were cultured in a test tube with PDA medium at 20 °C in darkness. We observed them after culturing for 7 d. (**B**) Colony morphology: The colonies were cultured on PDA medium in darkness for 21 d at 20 °C. We observed them after exposure to light/dark (12/12 h) condition for 4 d. (**C**) The hydrophobicity of mycelia. (**D**) The hydrophobicity of conidia. (**E**) Growth rates. (**F**) Conidia production. Error bars represent the standard deviation (SD) of three replicates. Different letters above the bars represent significant differences (*p* < 0.05).

**Figure 3 ijms-24-04586-f003:**
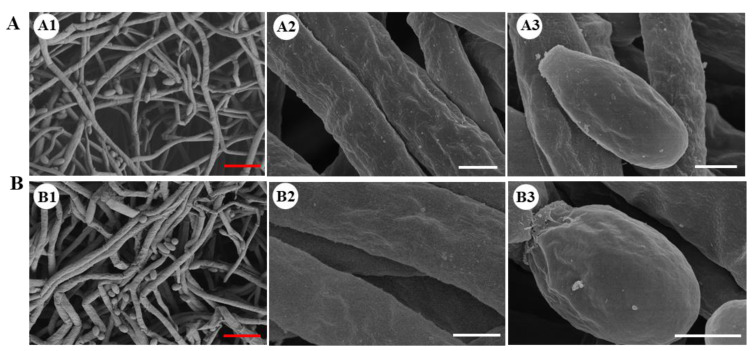
Hyphae and conidia of WT (**A**) and Δ*Cmhyd4* (**B**) strains observed by SEM. A1 and B1 showed the overview of the hyphae and conidia; A2 and B2 showed the surface of the hyphae; A3 and B3 showed the surface of the conidia. Bars: A1, B1 = 1 μm; others = 10 μm.

**Figure 4 ijms-24-04586-f004:**
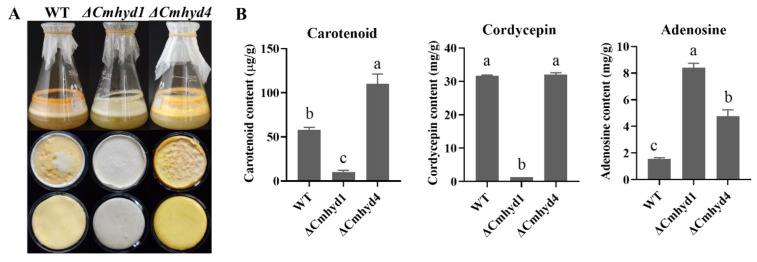
The morphology and secondary metabolite production of WT, Δ*Cmhyd1* and *ΔCmhyd4* strains by two-stage cultivation. (**A**) Fermentation broth, upper and reverse sides of the vela from the WT, Δ*Cmhyd1* and Δ*Cmhyd4* strains after two-stage cultivation for 18 d; (**B**) Contents of carotenoid in the vela, cordycepin, and adenosine in fermentation broth for the WT, Δ*Cmhyd1*, and Δ*Cmhyd4* strains. Error bars represent the SD of three biological replicates with three technical replicates. Different letters above the bars represent significant differences (*p* < 0.05).

**Figure 5 ijms-24-04586-f005:**
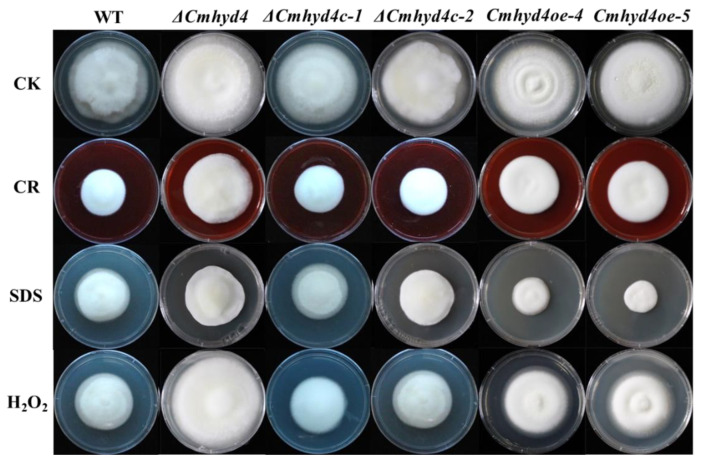
The stress response of mycelia in the WT and mutant strains. As the control, CK was inoculated on PDA medium. CR, SDS, and H_2_O_2_ represent the PDA with 10% congo red, 0.1% SDS, and 0.075 mM H_2_O_2_. All the strains were cultured at 20 °C for 21 d in darkness and then photographed. The experiments were repeated three times with similar results.

**Figure 6 ijms-24-04586-f006:**
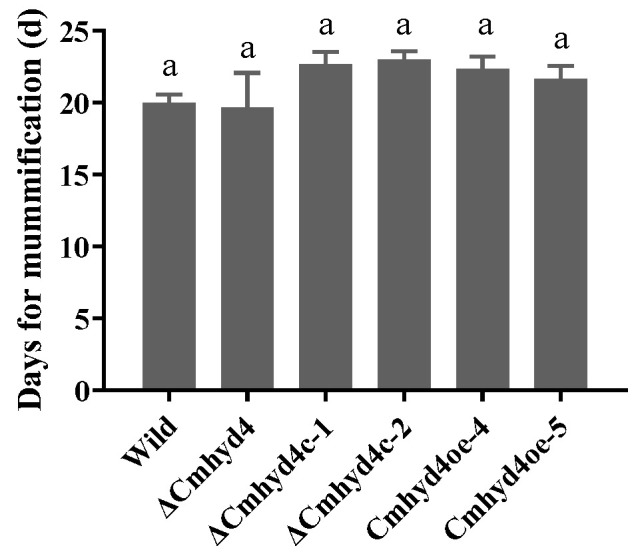
Conidial infecting ability of WT and mutant strains. Error bars represent the SD of three biological replicates with three technical replicates. The same “a” above the bars indicates that there is no significant difference (*p* < 0.05).

**Figure 7 ijms-24-04586-f007:**
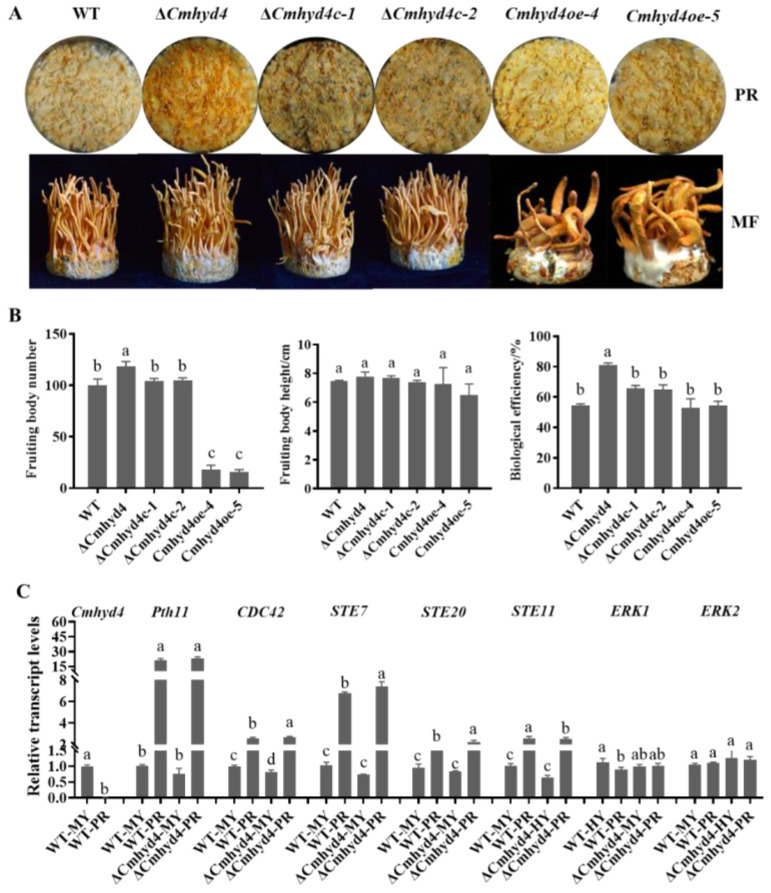
*Cmhyd4* negatively regulated fruiting body development in *Cordyceps militaris*. (**A**) Fruiting body formation and development in WT and mutants—PR: primordium formation (stroma < 1 cm); MF: mature fruiting body. (**B**) Mushroom yield of WT and mutants. After inoculation for 45 d, fruiting bodies were harvested and evaluated. (**C**) The relative transcription levels of *Cmhyd4* and fruiting-body-development-related genes at the primordial stage. Error bars represent the SD of three biological replicates with three technical replicates. The different letters over the histogram represent significant differences at *p* < 0.05.

## Data Availability

Data is contained within the article or Appendix A.

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
