# Peer review of "Hydrophobin Gene Cmhyd4 Negatively Regulates Fruiting Body Development in Edible Fungi Cordyceps militaris"

_ijms, 2023, doi:10.3390/ijms24054586_

Round 1

Reviewer 1 Report

Researches of Cordyceps militaris provide great values for the potential applications. In this study, authors described a negative regulation of Cmhyd4 in the fruiting body formation of Cordyceps militaris, and the results were well present. However, several issues need to be addressed. 

(1) Authors should supply the importance of Cordyceps militaris in the part of Introduction.

(2) In line 66, authors described that four hydrophobin-encoding genes are found in C. militaris, the reference or related data should be provided.

(3) In Figure 2, authors showed increased Conidium production for ΔCmhyd4, while in the SEM analysis (figure 3, A2 and B2), it seems that contrary result is present.

(4) Why the strain color of ΔCmhyd4 from Figure 2b was different from Figure 5? Otherwise, In Figure 5, the statistical analysis of the colony diameter should be provided. Some descriptions, like ΔCmhyd4, the Δ should be present as Δ.  

Reviewer 2 Report

This research reported  a regulatory gene which can regulate fruting body development in the famous fungus Cordyceps militaris. This results are interesting and may have utilization potencial in the fruting body production. The experiments are sound and the results are reliable. The only drawback is that there are still some problems in writting. Cordycepic acid (Line 61) should be better replaced with mannitol for that it is the wrong name of mannitol. "In mushroom breeding" (Lines 26-27, 371-372) should be better replaced with C. militaris for that the research have just studied the fungus C. militaris. Some sentence are worth reconsidering, such as lines 22-23, 219-222.

Reviewer 3 Report

The manuscript reports on the targeted gene deletion (and subsequent genetic complementation) of the Cmhyd4 gene encoding a putative hydrophobin in the medicinal fungus Cordyceps militaris.  Loss of the gene has a number of effects on this fungus, including changing levels of metabolites, stress resistance and the properties of the fruiting bodies.  This work would likely appeal to those interested in hydrophobins and/or development if macroscopic structures in the fungi.  But if one criticism could be raised, it is hard to see a broader appeal in knocking out a single gene and finding some phenotypes, as rather than incremental advance on understanding these aspects of fungal biology.

Minor typographical points are as follows.

Hyrophobins are extensively studied in the fungi, so picking representative studies to mention in the Introduction is difficult.  Perhaps it is therefore worth focusing on a smaller group, e.g. what is known in the close relatives of C. militaris, like other Hypocreales species.

Line 34: can delete ‘higher’ or replace with something else, or could be ‘filamentous fungi in the Dikarya lineage’.

Lines 50 and 66: ‘proved’ could be ‘found’.

Line 64: ‘genes’ for gene’.

Line 67: change ‘4’ to ‘the four’.

Line 72: duplicate text so delete ‘using edible fungi C. militaris as a model system.’

Lines 81, 305, 311 and 383: ‘Southern’ capital ‘S’.

Figure 1: for logic in order of presentation, it would make more sense to flip panels C and D, i.e. Southern before transcript levels.  For the Southern blot, there is no mention of which enzymes were used: guessing a double digest with BamHI and HindIII?

Line 144: overexpression strain rather than deletion strain.

Line 160: change ‘at’ to ‘in’.

Line 212: delete ‘While’.

Line 248: ‘While some functional’ could be ‘Other’.

Line 284: add italics to the species name.

Line 350: add a word, e.g. ‘described method [19]’.

Lines 204-205 has three technical replicates while line 363 has two technical replicates.
